# Potential Increased Risk of Trisomy 18 Observed After a Fertilizer Warehouse Fire in Brazos County and TX

**DOI:** 10.3390/ijerph17072561

**Published:** 2020-04-08

**Authors:** Xiaohui Xu, Xiao Zhang, JeongWon Han, Yau Adamu, Bangning Zhang

**Affiliations:** Department of Epidemiology and Biostatistics, School of Public Health, Texas A&M University, 212 12 Adriance Lab Road, College Station, TX 77843, USA; zhangxgz@tamu.edu (X.Z.); jwhan17@gmail.com (J.H.); adam_5336@tamu.edu (Y.A.); bzhang1s@exchange.tamu.edu (B.Z.)

**Keywords:** trisomy, chromosomal anomalies, birth defects, fertilizer, fire

## Abstract

Background: In this paper, we aimed to investigate the potential impacts of a fire accident in a fertilizer warehouse on chromosomal anomalies, including Trisomy 21 (T21) and Trisomy (T18) among pregnancies in Brazos County, Texas. We conducted an observational study in Brazos County, TX, with all patients of T18 and T21 cases in the live births in Brazos County between 2005–2014. The prevalence of T18 and T21 before, during, and after the accident in Brazos County were calculated and compared. The Standardized Morbidity Ratio (SMR) was applied to compare the prevalence of T18 and T21 in Brazos County to the statewide prevalence in Texas after adjusting for maternal race and age. Compared with statewide risk, the risk of T18 during the impacted years in Brazos county was found to be significantly higher (SMR = 5.0, 95% Confidence Interval(CI): 2.19–9.89), while there was no significant difference before (SMR = 0.77, 0.13–2.54) and after the accident (SMR = 0.71, 0.12–2.36). However, the prevalence of T21 during the impacted years was not significantly different from those before or after the accident. This study conclusively suggests that this fertilizer fire may be related to the increased prevalence of T18 in Brazos County, though the findings warrant further investigation.

## 1. Introduction

On 30 July 2009, a fire occurred in a fertilizer warehouse located in Bryan, Texas, which dealt with agricultural fertilizers, including E-2 ammonium nitrate.The fire was left to burn itself out due to dangers of chemical exposures, and formed an extensive amount of flames and smoke which spread to 60 miles away from the facility. Although no explosion occurred, over 70,000 residents were immediately evacuated due to the severity of the potential explosion by ammonium nitrate (NH_4_NO_3_). The facility stored fertilizer-grade ammonium nitrate (FGAN) and blended it with other materials to create fertilizer [1]. Although we could not ascertain the inventory immediately before the warehouse fire incident, the website of the supplying company indicated that the company produced “a variety of agrochemical and industrial products, including regular nitric acid and concentrated nitric acid, mixed (nitrating) acids, sulfuric acid, and both agricultural- and industrial-grade ammonium nitrate”. The specific agricultural products listed on the company’s website include high-density ammonium nitrate (AN) and ammonia, industrial and mining products, specialty nitric acid blends with strengths ranging from 56% to 84%, concentrated nitric acid, sulfuric acid, mixed acid, low-density ammonium nitrate, ammonium nitrate solutions, and ammonia.

Furthermore, the available literature showed that major reported products of decomposition of FGAN-based fertilizers under high temperature included: nitrogen (N_2_): 19%–26%, water vapour (H_2_O): 45%–65%, nitrous oxide (N_2_O): 7%–20%, nitrogen oxides (NOx), and ammonia (NH_3_): 0%–9%, respectively, hydrogen chloride (HCl) and hydrogen fluoride (HF): 0.5–10%, respectively, ammonium chloride (NH_4_Cl): 0%–7%, and chlorine (Cl_2_): 2%–6% [2,3,4]. Exposure to those products of decomposition subsequently generated chemicals which may have had acute and chronic consequences. During the accident, no physical injuries or fatal cases were reported, while 10 individuals were admitted to St. Joseph Regional Health Center’s emergency room and College Station Medical Center due to respiratory symptoms caused by smoke inhalation. Except for those reported acute health outcomes, long-term health impacts, including pregnancy/birth outcomes associated with the fire accident were public concerns, but no study has investigated it yet. Trisomy 21 and 18 (T21 and T18), known as Down Syndrome and Edward Syndrome, are the most common autosomal trisomies, which may have an association with environmental exposures [5]. If an infant is born with three 21 chromosomes, rather than the normal pair, then the infant would have T21. As for the T18, there are three types of T18, including full, mosaic, and partial trisomy forms, where only a segment of the chromosome 18 long arm is present in triplicate [6]. The full T18 is the most common form, accounting for about 94% affected cases [7]. The prevalence of T18 in live births ranges from 1/3600 to 1/10,000, which is still an extremely rare defect [8,9]. As fetuses with T18 are more likely to die during embryonic and fetal life, the prevalence of T18 is expected to be much higher than the live birth prevalence [10,11,12]. Although the etiology of chromosomal anomalies has not been well understood, epidemiological studies have suggested the relationship between environmental exposures and human aneuploidy. Numerous studies indicated that exposure to agrochemicals is associated with human aneuploidy. For example, exposure to organochlorines has been linked to sperm aneuploidy in men from the general population [13,14,15]. Similarly, exposure to organophosphate insecticides (OP) [16], air pollutants [17,18], and pyrethroids [19,20] were associated with chromosomal abnormalities in human reproductive cells. Furthermore, several studies demonstrated chromosomal abnormalities and other genetic effects associated with exposure to agrochemical fertilizers in fertilizer factory workers [21,22] and in exposed plant systems [23]. The products of fertilizer decomposition could also generate toxic products. For example, it has also been reported that heterogeneous reactions of NH_3_ with secondary organic aerosols (SOA) generates nitrogen-containing compounds (NOC) and nitrogen-associated ambient particulate matter [24,25]. Additionally, the NOx produced from fertilizer decomposition favors nitration reaction in smoke plumes and produces nitrophenols [26,27,28].

In this study, we investigated the potential impacts of the fire accident on chromosomal anomalies, including T21 and T18, among pregnancies in Brazos County, Texas. Specifically, we examined the prevalence of T21 and T18 before the accident (2005–2008), during the impacted years of the accident (2009–2010), and after the accident (2011–2014), respectively. Additionally, we compared the risk of T21 and T18 in Brazos County to the statewide risk of Texas among these three periods of time after adjusting for maternal race and age.

## 2. Materials and Methods

### 2.1. Study Population

Brazos County is in Texas, which had a population of over 190,000 in 2010 (US Census Bureau). The Bryan College Station is referred to as the metropolitan area in the county, which is the 16th largest metropolitan area in Texas. In this analysis, all live births between 2005–2014 from the residents in Brazos County, Texas (TX) were selected from the Texas birth certificate and surveillance data, which was obtained from the Center of Health Statistics [29].The number of live births by maternal age (<30, and 30+ years old) and race/ethnicity (non-Hispanic White, non-Hispanic black, and Hispanic) from 2005–2008 (before the accident), 2009–2010 (the impacted years), and 2011–2014 (after the accident) was obtained to examine the impacts of the fire accident on the prevalence of chromosomal anomalies, including T21 and T18. Studies have suggested that ethnic differences may influence the prevalence of Down syndrome [30], where four maternal race groups, including African American, Mexican American, non-Hispanic White, and Other were included in the analyses. We then classified our study population into these four race groups, based on previous studies.

### 2.2. Ascertainment of T18 and T21

All T18 and T21 cases in the live births in Brazos County between 2005–2014 were identified through the Texas Birth Defects Registry, which was established in 1993 to identify the patterns of birth defects in Texas through an operation of a population-based, active surveillance system. The information on the number of T18 and T21 cases in Brazos County and the statewide prevalence of T18 and T21 by maternal age and race/ethnicity for the selected periods of time was obtained through the Center of Health Statistics.

### 2.3. Statistical Analysis

As the fire accident happened on 30 July 2009, for analysis purposes, we defined the years of 2005–2008 as the period before the accident, 2009–2010 as the impacted years of the accident, and 2011–2014 as the period after the accident. We assumed that the four-year periods before and after the accident could well-represent the natural prevalence of the disease when there was no similar accident present. We calculated the annual prevalence of T18 and T21 between 2005–2014 in Brazos County, Texas and examined whether the prevalence of T18 and T21 during the impacted years were significantly different from those in other periods of time. Chi-square tests were performed to test the difference. Additionally, a Standardized Morbidity Ratio (SMR) was used to compare the risks of T18 and T21 in Brazos County with the statewide rate of Texas in the same period after adjusting for maternal race and age. Specifically, SMR was defined as the ratio of observed and expected number of cases:SMR = (Observed number of cases)/(Expected number of cases),(1)
where the expected number of cases was estimated by assuming the risk of T18 and T21 in Brazos County to be the same as the statewide risk of Texas. For statistical analysis, the observed number of cases was considered as a random variable with a Poisson distribution, while the expected number of cases was an invariant. The mid-P exact test was used as the statistical significance test between the observed and expected number of cases and as the method for 95% confidence interval (CI) estimation [31]. To interpret the CI of SMR, the inclusion of “1” simply indicates that the observed and expected cases are not significantly different. All data analyses were performed using OpenEpi, Version 3.01 (Copyright (c) 2003, =2008 Andrew G. Dean and Kevin M. Sullivan, Atlanta, GA, USA) [32].

## 3. Results

Table 1 shows the distribution of live births by maternal race and age in Brazos County before the accident (2005–2008), during the impacted years (2009–2010), and after the accident (2011–2014). The total number of live births per year ranged from 2467 to 2780 births between 2005–2014 and did not show any time trend. The distributions of live births by maternal age and race before, during, and after the accident were similar. Among them, 45–48% of live births were delivered by White mothers, and 31–35% of births were from Hispanic mothers. Approximately 30% of live births were delivered by mothers older than 30 years of age.

Figure 1 presents the annual prevalence of T18 between 2005–2014. Before the accident of 2005–2008, the prevalence of T18 each year was 0.0–0.4 cases per 1000 live births. During the impacted years of the accident of 2009–2010, the annual prevalence of T18 was 0.7–1.8 cases per 1000 live births. After the accident of 2011–2014, the annual prevalence of T18 ranged from 0.0 cases to 0.38 cases per 1000 live births. Figure 2 shows the prevalence of T18 by three time periods. It indicates that the prevalence during the impacted years (1.3 cases per 1000 live births) was significantly higher than those before and after the accident (both were 0.19 cases per 1000 live births, *p* < 0.05).

Figure 3 presents the annual prevalence of T21 between 2005–2014. Before the accident of 2005-2008, the prevalence of T21 each year was 0.8–2.1 cases per 1000 live births. During the impacted years of the accident of 2009–2010, the annual prevalence of T21 was 1.1–1.8 cases per 1000 live births. After the accident of 2011–2014, the annual prevalence of T21 ranged from 0.7 cases to 1.5 cases per 1000 live births. Figure 4 shows the prevalence of T21 by three periods of time. It indicates that the prevalence during the impacted years (1.4 cases per 1000 live births) was not significantly higher than those before and after the accident (1.3 and 1.2 cases per 10,000 live births, respectively).

Table 2 presents the SMRs comparing risks of T18 and T21 and their 95% CIs in Brazos County, TX as compared to those in the state of Texas by three time periods. As compared to the statewide risk, the risk of T18 in Brazos County was significantly higher during the impacted years (SMR = 5.00, 95% CI: 2.19–9.89), while there were no differences before (SMR = 0.77, 95% CI: 0.13–2.54) and after the fire accident (SMR = 0.71, 95% CI: 0.03–3.52) after controlling for maternal race and age. For T21, no higher risk was observed in Brazos County as compared to the state of Texas before the accident (SMR = 1.11, 95% CI: 0.63–1.83) and during the impacted years (SMR = 1.14, 95% CI: 0.53–2.17) after controlling for maternal age and race. After the accident, the risk of T21 in Brazos county was significantly lower than the state risk (SMR = 0.46, 95% CI: 0.19–0.97).

## 4. Discussion

This study sought to investigate the association between environmental exposures from a fertilizer plant fire and chromosomal anomalies in Brazos County, Texas. The findings suggested the prevalence of T18 was exceptionally higher during 2009–2010 (i.e., the impacted years) than those before and after the accident in Brazos County, TX. Because the demographic composition, including age and race, did not change much during the study period, and according to our knowledge, no other major environmental accidents or natural disasters happened during the study period, it is unlikely that the changes of prevalence were due to these factors in this area. Further, compared to the risk at the state of Texas, a higher risk of T18 in Brazos County was observed during 2009–2010, but was not observed before and after the accident after controlling for maternal race and age. This result provided additional evidence to support the potential impact of the accident on risk of T18 among the residents in the county. However, the warehouse fire seemed to have had little impact on the prevalence of T21, as the prevalence before, during, and after the accident was about the same. Besides, as compared to the risk at the state level, no higher risk of T21 in Brazos County was also observed in any of the three periods. In addition, the Texas Birth Defects Registry was established in 1993 by the Texas Birth Defects Act and has been in operation since 1994. Statewide data became available in 1999. Thus, this system has already been operating for many years in the study area, and no changes about this system were reported during the study period [33].

Although we could not find animal studies to support our suggested association between chemical fertilizers and chromosomal abnormalities, cytomorphologic and genotoxic effects, including chromosomal abnormalities associated with chemical fertilizers have been reported in several plant and epidemiological studies. The use of the plant system to investigate cytological aberrations has been considered an excellent monitoring system for the detection of possible genetic hazards of environmental chemicals for several decades [34,35,36]. Kumar and Sana demonstrated various chromosomal abnormalities in plants after exposure to high concentrations of fertilizers [37]. Similarly, Verma et al. investigated the genotoxic effects of nitrogen fertilizers in plants and concluded that nitrogen fertilizer induced chromosomal aberrations, including fragmentation, bridges, and disorientations [23]. The effects of fertilizers on chromosomal behaviors using plant models have also been reported in other studies [38,39]. Genotoxic and chromosomal abnormality endpoints from plant studies remains relevant for toxicity and risk assessment of agrochemicals, including fertilizer [23]. Therefore, our findings may suggest the need to investigate the cytological aberrations and other mutagenic effects of fertilizers using mammalian experimental models. Furthermore, epidemiological studies demonstrated a significantly increased frequency of sister-chromatid exchange, chromosomal aberrations, and micronuclei among the fertilizer factory workers exposed to air pollution from phosphate fertilizers. Although fluorine (HF and SiF_4_) was the main air pollutant in the investigated workplace, the workplace also contained ammonia (NH_3_), sulfur dioxide (SO_2_), and floating dust in small amounts, and their possible contribution towards the observed effects has been suggested [21,22]. Moreover, it could be plausibly proposed that these products of fertilizer decomposition and subsequent gasses could have been mutagenic agents leading to chromosomal abnormalities, depending on the composition of the fertilizer and condition of decomposition. For example, Grant et al. suggested that N_2_, a product from decomposition of ammonium nitrate by high temperature (170–280 °C), has been associated with a significant increase of chromosomal impairment [40,41,42]. Therefore, well-designed experimental studies are needed to investigate the association between products of fertilizer decomposition and chromosomal abnormalities in animals.

T18 and T21 are the most common autosomal trisomies among live births, and maternal-origin nondisjunction errors of the meiotic divisions during reproductive cell development have been reported to be the main cause [43]. Unlike T21 and all other human trisomies which mostly result from nondisjunction error in maternal meiosis phase I, T18 shows a higher frequency in maternal meiosis phase II [44,45]. In the development of ovum egg in humans, the primary oocyte undergoes meiosis phase I during embryonic development, and halts in the stage of prophase I until puberty. During each menstrual cycle, the primary oocyte completes meiosis I the day before ovulation and is developed into the secondary oocyte. The secondary oocyte initiates meiosis II immediately after meiosis I, and halts at the stage of metaphase II and will not continue without fertilization. At fertilization, meiosis II is completed [46]. The possible longer duration of meiosis II may have increased the chance of being exposed to hazardous pollutants that could potentially impact meiotic divisions [47], which may have accounted for the observed increased T18 incidence, not T21. Besides, studies reported that two conditions were required for T21 nondisjunction—first, the initiation of a vulnerability bivalent factor in prophase I, followed by abnormal processing of the bivalent at metaphase I or II [48,49]. It would be rare to meet both requirements. This suggests that exposure that occurs during metaphase II would be more likely to result into T18, since the first condition for T21 had not been initiated. This may further explain why there was an increase in T18, not T21.

Since nondisjunction errors during meiosis II are likely to be the cause of T18, the live births in 2009 were less likely to be impacted by the fire accident because the conception date of these births were before the accident (30 July 2009). In contrast, quite a few live births in 2010 took place near or after the time of the event. Therefore, the live births in 2010 were more likely to have been impacted by the fire accident than those in 2009. Due to the fact that pregnancies with T18 have a good chance of fetal loss and stillbirth, the actual T18 incidence in Brazos County was expected to be higher than that observed. However, the actual number of trisomy-affected conceptions was unlikely to be obtained, due to early pregnancy loss that occurs prior to a diagnosis. Studies found that approximately 22%–30% of all conceptions are lost around the time of the normal menstrual period [50]. A retrospective medical chart review that contains information of all recorded conceptions and spontaneous abortions may enable us to obtain more precise estimates of the potential impact of this accident on pregnancies in the county.

Limitations due to the availability of data should be noted. First, paternal age has been suggested to be associated with T18 [51,52]. Unfortunately, this information is not available in the data, and therefore we were not able to adjust for it. Second, in addition to T18 and T21, Trisomy 13 (T13, Patau’s syndrome) is another chromosomal abnormality that has been widely studied, but we did not include it in the analysis because of missing data. Third, we were not able to obtain information about the family history of prior genetic diseases among pregnant women, which is also a strong factor that may have affected the observed number of cases that should have been adjusted for. We also had live birth data collected by calendar year only, and therefore were unable to divide the analysis periods by the exact accident date, which may have attenuated the risk estimate. Lastly, we included all the live births in Brazos County, TX during the impacted years, and included a 95% confidence interval for SMR. Thus, the observed higher risk was unlikely to be due to random variations.

## 5. Conclusions

In conclusion, the fertilizer plant fire accident may have been associated with an increased risk of T18 in Brazos County, TX. Future well-designed studies should be conducted to evaluate the possible chromosomal effects of fertilizer and its products of decomposition under different conditions. Safety training and emergency responses should be enhanced to prevent similar accidents from happening in future.

## Figures and Tables

**Figure 1 ijerph-17-02561-f001:**
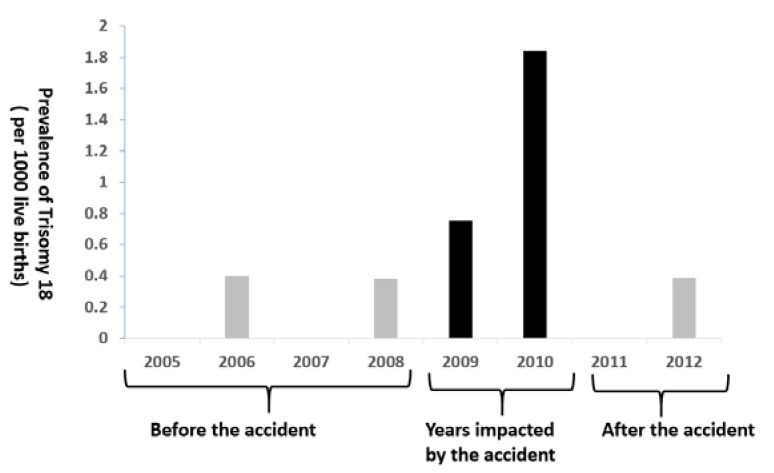
Annual prevalence of T18 between 2005–2012 in Brazos County, Texas.

**Figure 2 ijerph-17-02561-f002:**
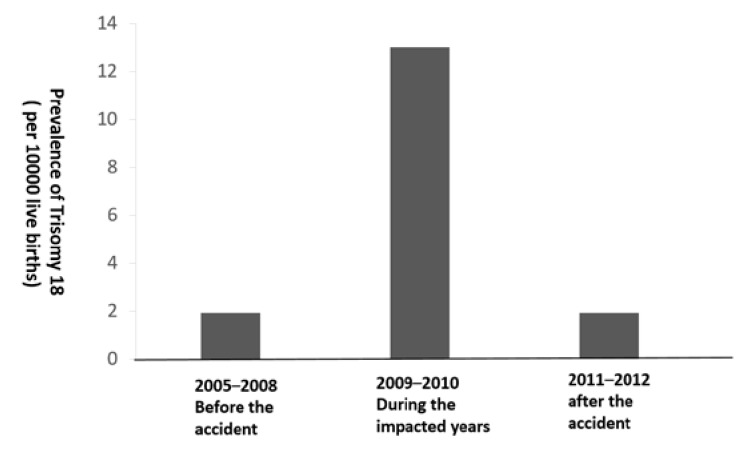
Prevalence of Trisomy 18 before, during, and after the fire accidents in Brazos County, Texas.

**Figure 3 ijerph-17-02561-f003:**
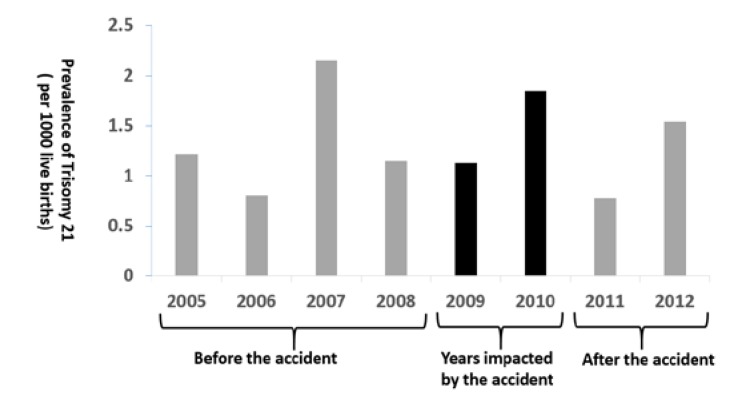
Prevalence of Trisomy 21 in 2005–2012 in Brazos County, Texas.

**Figure 4 ijerph-17-02561-f004:**
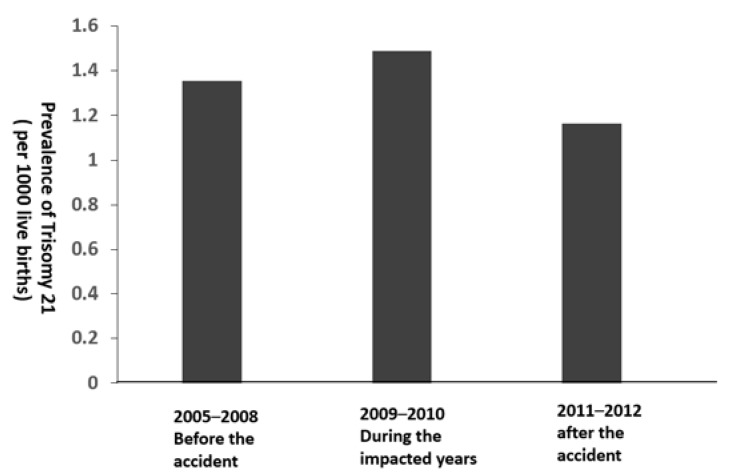
Prevalence of Trisomy 21 before, during, and after the fire accident in Brazos County, Texas.

**Table 1 ijerph-17-02561-t001:** The distribution of live births by maternal race and age in Brazos County, Texas, 2005–2014.

Maternal Race	Maternal Age (Years)	2005–2008 (Before the Accident)		2009–2010 (During the Impacted Years)		2011–2014 (After the Accident)	
		N	%	N	%	N	%
White	<30	3040	29.4	1652	30.8	3014	28.4
	30+	1622	15.7	887	16.5	1979	18.6
Black	<30	1160	11.2	574	10.7	1092	10.3
	30+	198	1.9	106	2.0	260	2.4
Hispanic	<30	2830	27.3	1339	24.9	2447	23.0
	30+	861	8.3	481	9.0	1018	9.6
Other	<30	272	2.6	128	2.4	366	3.4
	30+	368	3.6	205	3.8	448	4.2
Total		10351	100	5372	100	10624	100

**Table 2 ijerph-17-02561-t002:** Standardized Morbidity Ratios (SMRs) of trisomy 18 and 21 in Brazos County compared with the state-wide risk of Texas.

Chromosomal Birth Defects	Time Periods	Observed Number of Cases	Expected # of Cases	SMR	95% CI ^a^		*p*-Value ^a^
T18 (Edwards Syndrome)	2005–2008	2	2.6	0.77	0.13	2.54	0.71
	2009–2010	7	1.4	5.00	2.19	9.89	<0.001
	2011–2014	2	2.8	0.71	0.12	2.36	0.63
T21 (Down Syndrome)	2005–2008	14	12.6	1.11	0.63	1.82	0.69
	2009–2010	8	7.0	1.14	0.53	2.17	0.71
	2011–2014	14	26.3	0.53	0.30	0.87	0.016

^a^ CI and *p*-values were calculated based on the mid-P exact test.

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
