# Peer review of "Potential Increased Risk of Trisomy 18 Observed After a Fertilizer Warehouse Fire in Brazos County and TX"

_ijerph, 2020, doi:10.3390/ijerph17072561_

Round 1

Reviewer 1 Report

This paper performed epidemiologic analysis of birth defects caused by the Texas fertilizer fire in 2009. The data is comprised of the total of 26347 live births, including 11 cases of T18. Because the sample size is highly limited, consideration would be given to the following points.

  1. Please provide a list of cases of T18 and T21, which includes maternal age and month/year of birth.
  2. Periods which divided date of birth of patients are not clear. They can be divided by these three periods; before July 2009, during 12/24 months after fire outbreak, after August 2011/2012.
  3. L115; The distributions of live births by maternal age and race before, during and after the accident were similar. >>>>>Please show data for this statement.
  4. Why does this study classify patients into four maternal race groups? Does any study on T21 and T18 show significant difference between maternal races? Is the risk influenced by paternal race?
  5. If available, the number of miscarriages would be added.
  6. Please indicate ‘Brazos County’ in Figure 1.

Author Response

Point 1: Please provide a list of cases of T18 and T21, which includes maternal age and month/year of birth.

Response 1:

The reviewer’s recommendation is very critical and greatly appreciated for the clarification of our study. The list of cases includes maternal age and month/year of birth can provide a very clear vision for audiences to see the overall picture.   

Unfortunately, due to small number of cases and the confidential issue, the information at the individual level is not allowed to be shared or published.  

Point 2: Periods which divided date of birth of patients are not clear. They can be divided by these three periods; before July 2009, during 12/24 months after fire outbreak, after August 2011/2012.

Response 2:

We greatly appreciate the reviewer’s comments on the division dates. However, the observed number of T18 and T21 cases was only available by year, according to the data from the Center of Health Statistics (http://healthdata.dshs.texas.gov/VitalStatistics/Birth). Thus, we are not able to divide our data into the suggested three periods based on both year and month. We added this to our limitation (Line 222-223).

Point 3: L115; The distributions of live births by maternal age and race before, during and after the accident were similar. >>>>>Please show data for this statement.

Response 3:

We thank the reviewer for this comment. The distributions of live births by maternal age and race before, during and after the accident were similar (before (29.4%), during (30.8%) and after (28.4%) the accident among <30 years old White mothers; before (8.3%), during (9.0%), and after (9.6%) the accident among +30 years old Black mothers). There was no statistical significance observed.  

Point 4: Why does this study classify patients into four maternal race groups? Does any study on T21 and T18 show significant difference between maternal races? Is the risk influenced by paternal race?

Response 4:

The reviewer’s questions are critical to our study regarding the classification of race groups.  We should have put more information on this since the classification of race groups should base on scientific reasoning to improve the clarification of our study. Studies have suggested that ethnic differences may influence the prevalence of Down syndrome (B Khoshnood et al. 2015; Canfield, Mark A et al. 2015), where four maternal race groups including African American, Mexican American, non-Hispanic White, and Other were included in the analyses. We then based on previous studies classified our study population into these four race groups. This information has been added in the revised manuscript (Line 104 through Line 108) and highlighted in red.

The reviewer’s question regarding the association between paternal race and risk of T18 and T21 is also very important, as there is biological plausibility that race from both parents would have an impact on the risk of infant’s trisomy. As far as we know, there are very limited studies on paternal race and T21 and T18. However, the paternal age is suggested to be associated with the T18 (De Sourza E et al. 2010; Dundar, Munis et al. 2011). Unfortunately, this information is not available in the publicly available data from the Texas Department of State Health Services; we have added this as a limitation of our study in the Discussion Section (Line 215-217), which is highlighted in red.  

Reference:

Canfield, Mark A et al. “The association between race/ethnicity and major birth defects in the United States, 1999-2007.” American journal of public health vol. 104,9 (2014): e14-23. doi:10.2105/AJPH.2014.302098

Khoshnood, B et al. “Ethnic differences in the impact of advanced maternal age on birth prevalence of Down syndrome.” American journal of public health vol. 90,11 (2000): 1778-81. doi:10.2105/ajph.90.11.1778

De Souza E, Morris JK, Case–control analysis of paternal age and trisomic anomaliesArchives of Disease in Childhood 2010;95:893-897.

Dundar, Munis et al. “Prediction, prevention and personalisation of medication for the prenatal period: genetic prenatal tests for both rare and common diseases.” The EPMA journal vol. 2,2 (2011): 181-95. doi:10.1007/s13167-011-0080-3

Point 5: If available, the number of miscarriages would be added.

Response 5:

We agree with the reviewer that adding this information could greatly improve our study. Unfortunately, this information is not available in the data from the DSHS.

Point 6: Please indicate ‘Brazos County’ in Figure 1.

Response 6:

The reviewer’s recommendation is highly appreciated, as this modification will make our figure more precise and clearer to the audiences.  Please see the updated Figure 1b.

Reviewer 2 Report

This study is designed to determine the possible mutagenic effects of fertilizer warehouse fires on pregnancy outcomes (trisomy 21 and trisomy 18) in Brazos county, Texas during the years 2009-2010. The data was collected from county health registers for trisomy 21 and trisomy 18 for comparisons of frequencies of these two abnormalities with the period  with no chemical disasters. The authors conclude that there may be an effect of fertilizer fire exposure on the increase frequency of trisomy 18.

General comments: 

The second paragraph in discussion (line 162-196) is too much speculative and is recommended to be deleted unless there is published support for the findings of this study.

Similarly, the third paragraph in discussion (line 197-225) is also very speculative with no evidence and is recommended to be deleted as well

The authors can use  animal studies data if available  on the effects of the specific fertilizer fire of the warehouse to support their conclusions. 

There is a significant difference in the number of live births between whites and Hispanics. This will create a bias in the calculation. Please explain.

Specific Comments:

Add the details of the fertilizer composition in the warehouse in the introduction and add evidence from the literature that this particular fertilizer was proven to have a significant impact on chromosomal abnormalities at level which the authors presumed in their study.

Line 63-64: chromosomal anomalies including T21 and T18... Did the authors noted any other chromosomal abnormalities beside T21 and T18? 

Author Response

Point 1: The second paragraph in discussion (line 162-196) is too much speculative and is recommended to be deleted unless there is published support for the findings of this study.

Response 1:

The reviewer’s suggestion is very important because deletion and modification of the suggested lines allow us to focus on discussing the possible effects of fertilizer and its compositions which would make our paper more scientifically appealing.

The suggested lines have been mostly deleted and the remaining have been modified or moved to Lines 37-53 of the revised manuscript.

Point 2: Similarly, the third paragraph in discussion (line 197-225) is also very speculative with no evidence and is recommended to be deleted as well

Response 2:

By following the reviewer’s suggestion, we deleted these sentences.

Point 3: The authors can use  animal studies data if available  on the effects of the specific fertilizer fire of the warehouse to support their conclusions.

Response 3:

The reviewer’s comment is very relevant and greatly appreciated for charging us to provide biological plausibility from the literature to support our conclusions. The response is provided in Lines 187 -214 in the revised manuscript.

Specifically, the following information has been added to the Discussion section:

“Although we could not find animal studies to support our suggested association between chemical fertilizers and chromosomal abnormalities, cytomorphologic and genotoxic effects including chromosomal abnormalities associated with chemical fertilizers have been reported in several epidemiological and plant studies. The use of plant system to investigate cytological aberrations have been considered an excellent monitoring system for the detection of possible genetic hazard of environmental chemicals for several decades (Grant, W. F. 1978; Nilan and Vig 1976; Leme et al., 2009). Kumar and Sana have demonstrated various chromosomal abnormalities after exposing plant system to varying concentrations of fertilizers (Kumar, G., & Naseem, S. 2011). Similarly, Verma et al. have investigated genotoxic effects of nitrogen fertilizers in plant and concluded that nitrogen-based fertilizer induces chromosomal aberrations including fragmentation, bridges and disorientations (Verma et al., 2016). The effects of fertilizers on chromosomal behaviors using plant models have been reported from other studies (El-Nahas 2000; Tabur & Oney, 2009). Genotoxic and chromosomal abnormality endpoints from plant studies remains relevant for toxicity and risk assessment of agrochemicals including fertilizer (Verma et al., 2017). Therefore, our findings may suggest the needs to investigate the cytological aberrations and other mutagenic effects of fertilizers using mammalian experimental models. Furthermore, few epidemiological studies demonstrated significantly increased frequency of sister-chromatid exchange, chromosomal aberrations and micronuclei and among fertilizer factory workers exposed to air pollution from phosphate fertilizers. Although fluorine (HF and SiF4) were the main air pollutants in the investigated workplace pollutants, the workplace pollutants contains ammonia (NH3), sulfur dioxide (SO2) and floating dust in small amounts and their possible contribution towards the observed effects has been suggested (Meng et al., 1997; 1995). ”

References:

Tabur, S., & Oney, S. (2009). Effect of artificial fertilizers on mitotic index and chromosome behaviour in Vicia hybrida L. Journal of Agricultural Research (03681157), 47(1).

El-Nahas, A. I. (2000). Mutagenic potential of imazethapyr herbicide (pursuit) on Vicia faba in the presence of urea fertilizer. Pak J Biol Sci, 3(5), 600-905.

Kumar, G., & Naseem, S. (2011). Mitoclastic responses of biofertilizer and chemical fertilizer in root meristems of Trigonella foenum graecum L. Chromosome Botany, 6(4), 121-124.

Verma, S., Arora, K., & Srivastava, A. (2016). Monitoring of genotoxic risks of nitrogen fertilizers by Allium cepa L. mitosis bioassay. Caryologia, 69(4), 343-350. https://doi.org/10.1007/s10661-017-5873-y

Meng, Z., Meng, H., & Cao, X. (1995). Sister-chromatid exchanges in lymphocytes of workers at a phosphate fertilizer factory. Mutation Research/Environmental Mutagenesis and Related Subjects, 334(2), 243-246.

Meng, Z., & Zhang, B. (1997). Chromosomal aberrations and micronuclei in lymphocytes of workers at a phosphate fertilizer factory. Mutation Research/Genetic Toxicology and

Environmental Mutagenesis, 393(3), 283-288.

Grant, W. F. (1978). Chromosome aberrations in plants as a monitoring system. Environmental Health Perspectives, 27, 37-43.

Grant, W. F. (1982). Chromosome aberration assays in Allium: A report of the US Environmental Protection Agency gene-tox program. Mutation Research/Reviews in Genetic Toxicology, 99(3), 273-291.

Leme, D. M., & Marin-Morales, M. A. (2009). Allium cepa test in environmental monitoring: a review on its application. Mutation Research/Reviews in Mutation Research, 682(1), 71-81.

Nilan, R. A., and Vig, B. K. Plant test systems for detection of chemical mutagens. In: Chemical Mutagens; Principles and Methods for Their Detection. Vol. 4, A. Hollaender, Ed., Plenum Press, New York, 1976, 143.

Point 4: There is a significant difference in the number of live births between whites and Hispanics. This will create a bias in the calculation. Please explain.

Response 4:

During the impacted years between 2009 and 2010, the number of live births between Whites and Hispanics did not show significant difference based on t-test (p-value=0.5955, t-statistics=0.6255). Moreover, there was no significant difference among the number of live births among four race groups (p-value=0.0903, F-statistics=5.8570). Thus, this would be very unlikely to create bias in the calculation.

Point 5: Add the details of the fertilizer composition in the warehouse in the introduction and add evidence from the literature that this particular fertilizer was proven to have a significant impact on chromosomal abnormalities at level which the authors presumed in their study.

Response 5:

We made the revision based on the reviewer’s suggestions.

The detailed response has been provided in Lines 37-53 and Lines 75-82 in the revised manuscript.

Specifically, the following details of the fertilizer composition in the warehouse have been added to the Introduction section:

“The facility stores Fertilizer-Grade Ammonium Nitrate (FGAN) and blends it with other materials to create fertilizer (Willey, R. 2017, 62p, 103p, 129p, 131p). Although we could not ascertain the inventory at the immediately before the warehouse fire incident, the website of the supplying company indicated that the company ‘produces a variety of agrochemical and industrial products including regular nitric acid and concentrated nitric acid, mixed (nitrating) acids, sulfuric acid, and both agricultural and industrial grade ammonium nitrate’. The specific Agricultural Products listed on the company’s website include High Density Ammonium Nitrate (AN) and Ammonia, Industrial & Mining Products, Specialty nitric acid blends with strengths ranging from 56 to 84%, Concentrated Nitric Acid, Sulfuric Acid, Mixed Acid, Low Density Ammonium Nitrate, Ammonium Nitrate Solutions and Ammonia.  (https://www.lsbindustries.com/edc). Furthermore, the   available literature showed that major reported products of decomposition of FGAN-based fertilizers under high temperature include: nitrogen (N2): 19-26 %, water vapour (H2O): 45-65%, nitrous oxide (N2O): 7-20%, nitrogen oxides (NOx), and ammonia (NH3): 0-9% respectively, hydrogen chloride (HCl) and hydrogen fluoride (HF): 0.5-10% respectively, ammonium chloride (NH4Cl): 0-7%, chlorine (Cl2): 2-6% (Christiansen et al., 1993; EFMA 2007; Török et al. 2015).”

Point 6: Line 63-64: chromosomal anomalies including T21 and T18... Did the authors noted any other chromosomal abnormalities beside T21 and T18?

Response 6:

In addition to T18 and T21, Trisomy 13 (T13, Patau’s syndrome) is another chromosomal abnormality that has been widely studied. However, the number of observed T13 cases is below 5 cases which was exempted by the DSHS due to privacy concern and so we were not able to perform analyses on T13. We have added this as a limitation of this study in the discussion (Lines 217 to 219).

Reviewer 3 Report

I was invited to revise the paper entitled "Increased Risk of Trisomy 18 Observed after a 1 Fertilizer Warehouse Fire in Brazos County, TX". The paper aimed to investigate the possible association of a fertilazire warehouse fire accident on the incidence of trisomy pregnancy. The topic is very interesting and the paper can improve knoledge about possible relevant associatiion between genetic diseases and environmental pollution. Despite this, I have some points to raise:

a. Introduction:

Background well presented. In Line 43 please cite articles reffered to the sentence "known as Down Syndrome and Edward Syndrome, 42 which may have an association with environmental exposures". 

b. Methods:

  • Authors did not reported which kind of chemical products were released in the environment. It is important because future study can investigate possible causal effect of them. In addition, Authors can discuss their results by the knowledge of this point. 
  • Authors reported that the accident happened in July 2009, so all deliveries performed prior this date, cannot be referred to the impacted years. Authors should add all deliveries performed prior the accident to the period befor the accident.
  • I suggest to perform a trend analysis in addition to the present analysis in order to evaluate the change over time of the prevalence of trisomies. The trend analysis, performed by standardized rate, can help to adjust the possible influence of age on the trisomy rates. From the analysis performed by Authors, we can't exclude the impact of maternal age on trisomy occurrence.
  • Authors are able to obtain information about family history of prior genetic disease among pregnant who delivered infants with trisomy?

c. Discussions

As previously reported, Authors should focus the discussion also on chemical products released in the environment. 

Author Response

Point 1: Background well presented. In Line 43 please cite articles referred to the sentence "known as Down Syndrome and Edward Syndrome, 42 which may have an association with environmental exposures".

Response 1:

We have added the citation for the sentence (Line 59).

Kihal-Talantikite, W., Zmirou-Navier, D., Padilla, C., & Deguen, S. (2017). Systematic literature review of reproductive outcome associated with residential proximity to polluted sites. International journal of health geographics, 16(1), 20.

Point 2: Methods: Authors did not reported which kind of chemical products were released in the environment. It is important because future study can investigate possible causal effect of them. In addition, Authors can discuss their results by the knowledge of this point.

Response 2:

Based on the reviewer’s suggestion, we added the response in Lines 47-53 of the revised manuscript, written in red.

Specifically, the added details are as below:

“The facility stores Fertilizer-Grade Ammonium Nitrate (FGAN) and blends it with other materials to create fertilizer (Willey, R. 2017, p. 62, 103, 129, 131). The available literature showed that major reported products of decomposition of FGAN-based fertilizers under high temperature include: nitrogen (N2): 19-26 %, water vapour (H2O): 45-65%, nitrous oxide (N2O): 7-20%, nitrogen oxides (NOx), and ammonia (NH3): 0-9% respectively, hydrogen chloride (HCl) and hydrogen fluoride (HF): 0.5-10% respectively, ammonium chloride (NH4Cl): 0-7%, chlorine (Cl2): 2-6% (Christiansen et al., 1993; EFMA 2007; Török et al. 2015).”

Authors reported that the accident happened in July 2009, so all deliveries performed prior this date, cannot be referred to the impacted years. Authors should add all deliveries performed prior the accident to the period before the accident.

The reviewer has mentioned a very important term of the period division for our study, which we greatly appreciate. The recommendation by the reviewer could improve the accuracy of our study; however, the publicly available data source from the Texas Department of State Health Services (DSHS) has limited information that they only provided the birth data by year not by year and month. Thus, we were unable to divide the period as suggested. We added this in our study’s limitations (Lines 222-223).

I suggest to perform a trend analysis in addition to the present analysis in order to evaluate the change over time of the prevalence of trisomies. The trend analysis, performed by standardized rate, can help to adjust the possible influence of age on the trisomy rates. From the analysis performed by Authors, we can't exclude the impact of maternal age on trisomy occurrence.

Authors are able to obtain information about family history of prior genetic disease among pregnant who delivered infants with trisomy?

Response 2:

In this analysis, we performed the indirect adjustment approach to estimate the Standardized Morbidity Ratios (SMRs) and the distributions of maternal age and race/ethnicity over three study periods were also very similar. Thus, the differences of the trisomy rates over times were less likely to be impacted by these factors. Additionally, the trisomy rates had a curved shape over time. A linear trend analysis would not provide us important insight for us.

We were not able to obtain the information about family history of prior genetic disease among pregnant women who delivered infants with T18 due to the availability of the data from the DSHS. This is a very important point recommended by reviewer and it has been added in the limitation part (Lines 219-222).

Point 3: Discussions: As previously reported, Authors should focus the discussion also on chemical products released in the environment

Response 3:

The reviewer’s concerns on focusing our discussion on chemical products released in the environment is very relevant and fruitful because it makes the manuscript more scientifically appealing.

This response is available in Lines 187-214 of the revised manuscript, written in red.

Specifically, the added details are as below:

“Although we could not find animal studies to support our suggested association between chemical fertilizers and chromosomal abnormalities, cytomorphologic and genotoxic effects including chromosomal abnormalities associated with chemical fertilizers have been reported in several plant and epidemiological studies. The used of plant system to investigate cytological aberrations have been considered an excellent monitoring system for the detection of possible genetic hazard of environmental chemicals for several decades (Grant, W. F. 1978; Nilan and Vig 1976; Leme et al., 2009). In plant, Kumar and Sana demonstrated various chromosomal abnormalities after exposure to high concentrations of fertilizers (Kumar, G., & Naseem, S. 2011).  Similarly, Verma et al. have investigated genotoxic effects of nitrogen fertilizers in plant and concluded that nitrogen fertilizer induces chromosomal aberrations including fragmentation, bridges and disorientations (Verma et al., 2016). More information on the effects of fertilizers on chromosomal behaviors using plant models have been reported from other studies (El-Nahas 2000; Tabur & Oney, 2009). Genotoxic and chromosomal abnormality endpoints from plant studies remains relevant for toxicity and risk assessment of agrochemicals including fertilizer (Verma et al., 2017). Therefore, our findings may suggest the needs to investigate the cytological aberrations and other mutagenic effects of fertilizers using mammalian experimental models. Furthermore, epidemiological studies demonstrated significance increased frequency of sister-chromatid exchange, chromosomal aberrations and micronuclei and among fertilizer factory workers exposed to air pollution from phosphate fertilizers. Although fluorine (HF and SiF4) are the main air pollutants in the investigated workplace, the workplace contains ammonia (NH3), sulfur dioxide (SO2) and floating dust in small amounts and their possible contribution towards the observed effects has been suggested (Meng et al., 1997; 1995). Moreover, it could be plausibly proposed that these products of fertilizer decomposition and subsequent gasses could be mutagenic agents leading to chromosomal abnormalities, depending on the composition of the fertilizer and condition of decomposition. For example, some Grant et al have  suggested that N2, a product from decomposition of ammonium nitrate by high temperature (170-280°C), has been associated with significant increase of chromosomal impairment (Grant et al. 1977; Laboureur et al. 2016; Musak et al. 2013). Therefore, well-designed experimental studies are needed to investigate association between products of fertilizer decomposition and chromosomal abnormalities in animals.”

References:

Tabur, S., & Oney, S. (2009). Effect of artificial fertilizers on mitotic index and chromosome behaviour in Vicia hybrida L. Journal of Agricultural Research (03681157), 47(1).

El-Nahas, A. I. (2000). Mutagenic potential of imazethapyr herbicide (pursuit) on Vicia faba in the presence of urea fertilizer. Pak J Biol Sci, 3(5), 600-905.

Kumar, G., & Naseem, S. (2011). Mitoclastic responses of biofertilizer and chemical fertilizer in root meristems of Trigonella foenum graecum L. Chromosome Botany, 6(4), 121-124.

Verma, S., Arora, K., & Srivastava, A. (2016). Monitoring of genotoxic risks of nitrogen fertilizers by Allium cepa L. mitosis bioassay. Caryologia, 69(4), 343-350. https://doi.org/10.1007/s10661-017-5873-y

Meng, Z., Meng, H., & Cao, X. (1995). Sister-chromatid exchanges in lymphocytes of workers at a phosphate fertilizer factory. Mutation Research/Environmental Mutagenesis and Related Subjects, 334(2), 243-246.

Meng, Z., & Zhang, B. (1997). Chromosomal aberrations and micronuclei in lymphocytes of workers at a phosphate fertilizer factory. Mutation Research/Genetic Toxicology and

Environmental Mutagenesis, 393(3), 283-288.

Grant, W. F. (1978). Chromosome aberrations in plants as a monitoring system. Environmental Health Perspectives, 27, 37-43.

Grant, W. F. (1982). Chromosome aberration assays in Allium: A report of the US Environmental Protection Agency gene-tox program. Mutation Research/Reviews in Genetic Toxicology, 99(3), 273-291.

Leme, D. M., & Marin-Morales, M. A. (2009). Allium cepa test in environmental monitoring: a review on its application. Mutation Research/Reviews in Mutation Research, 682(1), 71-81.

Nilan, R. A., and Vig, B. K. Plant test systems for detection of chemical mutagens. In: Chemical Mutagens; Principles and Methods for Their Detection. Vol. 4, A. Hollaender, Ed., Plenum Press, New York, 1976, 143.

Round 2

Reviewer 1 Report

The manuscript has partly improved; however, in addition to small sample size, data presented in the paper is highly limited.

  1. Because this study is critical in environmental problems, it is required to increase the sample size to provide a definitive conclusion. The title would cause a high impact on this issue without sufficient data. As described in Abstract and Conclusions, ‘further investigation’ and ‘well-designed studies’ are essential to avoid misleading interpretations of the data.
  2. Which stage is influenced by mutagens to cause T18 or T21? Major cause of trisomy would originate from oogenesis or spermatogenesis before fertilization. It is unlikely that births before December 2009 are affected by the accident. Periods before, during and after the accident may be reconsidered.

Author Response

Point 1: Because this study is critical in environmental problems, it is required to increase the sample size to provide a definitive conclusion. The title would cause a high impact on this issue without sufficient data. As described in Abstract and Conclusions, ‘further investigation’ and ‘well-designed studies’ are essential to avoid misleading interpretations of the data.

Response 1:

We greatly appreciate the reviewer’s comment and thus we have modified our title as “Potential Increased Risk of Trisomy 18 Observed after a Fertilizer Warehouse Fire in Brazos County, TX”.

We acknowledge that T18 and T21 are rare birth outcomes and increased sample size would improve the precision of our study estimates and provide a more definitive conclusion. However, we have included all the live births in Brazos County, TX during the impacted years, and we had included 95% Confidence Interval for SMR. Thus, the observed higher risk is unlikely due to random variations. We have added this as a limitation in the discussion section (Line255-258).

Point 2: Which stage is influenced by mutagens to cause T18 or T21? Major cause of trisomy would originate from oogenesis or spermatogenesis before fertilization. It is unlikely that births before December 2009 are affected by the accident. Periods before, during and after the accident may be reconsidered.

Response 2:

We greatly appreciate the reviewer’s comment. We have addressed the questions as below and added to the discussion Line 218-246:

T18 and T21 are the most common autosomal trisomies among live births, and maternal-origin nondisjunction errors of the meiotic divisions during reproductive cell development have been reported to be the main cause (Nothen et al. 1993). Unlike T21 and all other human trisomies which mostly result from nondisjunction error in maternal meiosis phase I, T18 shows a higher frequency in maternal meiosis phase II (Eggermann et al. 1996; Kupke and Muller 1989). In the development of ovum egg in humans, primary oocyte undergoes meiosis phase I during embryonic development and halts in the stage of prophase I until puberty. During each menstrual cycle, primary oocyte completes meiosis I the day before ovulation and is developed into secondary oocyte. The secondary oocyte initiates meiosis II immediately after meiosis I and halts at the stage of metaphase II and will not continue without fertilization, at fertilization, meiosis II completed (Hassold and Hunt 2001). The possible longer duration of meiosis II increases the chance of being exposed to hazardous pollutants that potentially impact meiotic divisions (Ghosh & Dey, 2013, chap.9, p149), which may account for the observed increased T18 incidence, not T21. Besides, studies reported that two conditions are required for T21 nondisjunction: first, the initiation of vulnerability bivalent factor in prophase I and followed by abnormal processing of the bivalent at metaphase I or II (Hassold and Sherman 2000; Oliver et al. 2008). It would be rare to meet both requirements. Suggesting that exposure that occurs during metaphase II would be more likely to result into T18, since the first condition for T21 has not been initiated. This may further explain why there was an increase in T18, not T21.

Since nondisjunction errors during meiosis II are likely the cause of T18, the live births in 2009 were less likely to be impacted by the fire accident because the conception date of these births were before the accident (July 30, 2009). In contrast, a good many live births in 2010 were around or after the event. Therefore, live births in 2010 were more likely to be impacted by the fire accident than those in 2009. Due to the fact that pregnancies with T18 have a good chance of fetal loss and stillbirth, the actual T18 incidence in Brazos County is expected to be higher than the observed.  However, the actual number of trisomy-affected conceptions is unlikely to be obtained due to early pregnancy loss that occurs prior to a diagnosis. Studies found approximately 22-30% of all conceptions are lost around the time of normal menstrual period (Wilcox et al. 1988). Retrospective medical chart review that contains information of all recorded conceptions and spontaneous abortions may enable us to obtain more precise estimates of the potential impact of this accident on pregnancies in the county.

References:

Nothen MM, Eggermann T, Erdmann J, Eiben B, Hofmann D, Propping P, Schwanitz G: Retrospective study of the parental origin of the extra chromosome in trisomy 18 (Edwards syndrome). Hum Genet 1993, 92(4):347-349.

Eggermann T, Nothen MM, Eiben B, Hofmann D, Hinkel K, Fimmers R, Schwanitz G: Trisomy of human chromosome 18: molecular studies on parental origin and cell stage of nondisjunction. Hum Genet 1996, 97(2):218-223.

Kupke KG, Muller U: Parental origin of the extra chromosome in trisomy 18. Am J Hum Genet 1989, 45(4):599-605.

Hassold T, Hunt P. 2001. To err (meiotically) is human: The genesis of human aneuploidy. Nature reviews Genetics 2:280-291.

Hassold T, Sherman S. 2000. Down syndrome: Genetic recombination and the origin of the extra chromosome 21. Clinical genetics 57:95-100.

Oliver TR, Feingold E, Yu K, Cheung V, Tinker S, Yadav-Shah M, et al. 2008. New insights into human nondisjunction of chromosome 21 in oocytes. PLoS genetics 4:e1000033.

Wilcox AJ, Weinberg CR, O'Connor JF, Baird DD, Schlatterer JP, Canfield RE, et al. 1988. Incidence of early loss of pregnancy. N Engl J Med 319:189-194.

Reviewer 2 Report

The authors have addressed the recommendations and comments in the manuscript.

Author Response

The authors have addressed the reviewer's comments. Thank you.

Reviewer 3 Report

Authors addressed all suggestions proposed. I'm very satisfied for all responses. The paper is actually improved and can be accepted for pubblication.

Author Response

(The authors gave the same response as above.)
